# Four New or Newly Recorded Species from Freshwater Habitats in Jiangxi Province, China

**DOI:** 10.3390/jof11010079

**Published:** 2025-01-19

**Authors:** Chen-Yu Xu, Hai-Yan Song, Jian-Ping Zhou, Zhi-Jun Zhai, Chao-Yu Cui, Dian-Ming Hu

**Affiliations:** 1Bioengineering and Technological Research Center for Edible and Medicinal Fungi, Jiangxi Agricultural University, Nanchang 330045, China; xuchenyu0227@163.com (C.-Y.X.); songhaiyan115@163.com (H.-Y.S.); zhoujp1991@126.com (J.-P.Z.); zhjzh002@163.com (Z.-J.Z.); 2Nanchang Key Laboratory of Edible and Medicinal Fungi, Jiangxi Agricultural University, Nanchang 330045, China; 3Jiangxi Key Laboratory for Excavation and Utilization of Agricultural Microorganisms, Jiangxi Agricultural University, Nanchang 330045, China; cycui@jxau.edu.cn; 4Key Laboratory of Crop Physiology, Ecology and Genetic Breeding, Jiangxi Agricultural University, Ministry of Education of the P.R. China, Nanchang 330029, China

**Keywords:** diversity, *Phaeoisaria*, *Pleurothecium*, *Pseudodactylaria*, taxonomy, three new species

## Abstract

Freshwater fungi consist of a highly diverse group of organisms in freshwater habitats worldwide. During a survey of fungal diversity in freshwater habitats across different regions of Jiangxi Province, China, four freshwater fungi were collected. To study their phylogenetic relationships, the internal transcribed spacer (ITS1-5.8S-ITS2), large subunit (28S, LSU), small subunit (18S, SSU), and RNA polymerase II subunit (*RPB2*) genes were selected for phylogenetic analyses. Based on morphology coupled with phylogenetic analysis, these strains were confirmed to belong to *Phaeoisaria*, *Pleurothecium*, and *Pseudodactylaria*. Among them, three fungi were confirmed as the new species, namely, *Pleurothecium lignicola* (Pleurotheciaceae), *Pseudodactylaria jiangxiensis* (*Pseudodactylariaceae*), and *Ps. lignicola* (Pseudodactylariaceae). One species was identified as *Phaeoisaria filiformis* (Pleurotheciaceae), a new record of this species in China. All species were compared with other similar species, and detailed descriptions, illustrations, and phylogenetic data were provided.

## 1. Introduction

Freshwater fungi refer to fungi that rely on freshwater for their whole or partial life cycle [1]. They are highly diverse and are distributed across thirteen phyla, namely, Aphelidiomycota, Ascomycota, Basidiomycota, Blastocladiomycota, Chytridiomycota, Entomophthoromycota, Monoblepharomycota, Mortierellomycota, Mucoromycota, Olpidiomycota, Rozellomycota, Sanchytriomycota, and Zoopagomycota [2,3,4,5]. Freshwater fungi are found in a wide range of substrates, such as leaf litter, plant debris, decaying wood, aquatic plants, insects, and soil [6]. They play an indispensable role as decomposers in freshwater ecosystems [7]. Consequently, the study of fungal biodiversity in freshwater habitats is of great significance, and this paper involves three genera across two families.

Pleurotheciaceae was established by Réblová et al. [8] and typified by the genus *Pleurothecium*. Xia et al. [9] assigned two *Rhexoacrodictys* species in Savoryellaceae with sequence data provided. However, Luo et al. [5] included species of Conioscyphales, Fuscosporellales, and Pleurotheciales in the phylogenetic analysis, and the result showed that these two *Rhexoacrodictys* species clustered within Pleurotheciaceae. This placement was ignored by Hyde et al. [10] while acknowledged by Shi et al. [11] and Yang et al. [12]. The phylogenetic result in this study (Figure 1) is consistent with Luo et al. [5], supporting the inclusion of *Rhexoacrodictys* within Pleurotheciaceae. At present, Pleurotheciaceae accommodates fourteen genera, namely, *Adelosphaeria*, *Anapleurothecium*, *Coleodictyospora*, *Dematipyriforma*, *Helicoascotaiwania*, *Melanotrigonum*, *Neomonodictys*, *Phaeoisaria*, *Phragmocephala*, *Pleurotheciella*, *Pleurothecium*, *Rhexoacrodictys*, *Saprodesmium*, and *Sterigmatobotrys* [5,10,11,12,13]. The majority of species within Pleurotheciaceae are saprobic on decaying wood and plant debris [14], whereas few species have been identified as opportunistic human pathogens [5,10,15].

*Phaeoisaria* was established by von Höhnel [16] to accommodate a hyphomycetous taxon collected from a bamboo substrate, with *P*. *bambusae* as the type species. This genus is characterized by indeterminate synnemata with compact and parallel adpressed conidiophores, polyblastic sympodially extending denticulate conidiogenous cells, and aseptate or septate ellipsoidal to obovoidal, fusiform-cylindrical conidia [5,8,16,17]. *P. filiformis* is the first and currently the only known sexual species in *Phaeoisaria* described by Luo et al. [5], which has immersed globose to elongate globose ascomata with a long, cylindrical, black neck, unitunicate, cylindrical asci with a small refractive apical apparatus and filiform, guttulate, hyaline conidia. To date, thirty-eight records are listed on Index Fungorum [18].

*Pleurothecium* was proposed by von Höhnel [19] and typified by *P*. *recurvatum*. Currently, eighteen species are accepted in *Pleurothecium*, of which only two species, *P*. *recurvatum* and *P*. *semifecundum*, have sexual morphs [18,20,21]. The asexual morph of *Pleurothecium* is characterized by brown, macronematous conidiophores, polyblastic, sympodially extended denticulate conidiogenous cells and solitary, hyaline or pigmented, unicellular or septate, cylindrical, ellipsoidal, fusiform or clavate conidia [5,20,21].

*Pseudodactylaria* was proposed by Crous et al. [22] based on the type species *Ps*. *xanthorrhoeae* to accommodate two dactylaria-like species, *Ps*. *hyalotunicata* and *Ps*. *xanthorrhoeae*, within the family Pseudodactylariaceae. Pseudodactylariaceae was established as a monotypic family in Pseudodactylariales by Crous et al. [22], which formed a distinct clade within the subclass Sordariomycetidae (Sordariomycetes). Hyde et al. [10] confirmed and admitted this treatment based on the phylogenetic analyses and divergence time estimates. To date, ten species are accepted in *Pseudodactylaria* [18].

In this study, we investigated freshwater fungi from Jiangxi Province, China, with the aim of identifying these fungi and clarifying their systematic placement. The research methodology employed a combination of morphological examination and multi-gene phylogenetic analysis. By meticulously observing the morphological characteristics of the fungal specimens and conducting rigorous phylogenetic analyses based on molecular data, we were able to accurately identify and classify the fungi. Utilizing this approach, we introduced three new species, namely, *Pleurothecium lignicola* sp. nov., *Pseudodactylaria jiangxiensis* sp. nov., and *Pseudodactylaria lignicola* sp. nov., as well as a new record species, *Phaeoisaria filiformis*.

## 2. Materials and Methods

### 2.1. Collection of Specimens, Morphology, and Isolation

Submerged decaying wood specimens were collected from freshwater habitats in Jiangxi Province, China. The specimens were placed in hermetic resealable bags, and the sampling information was recorded. They were then taken back to the laboratory. Upon arrival at the laboratory, the specimens were incubated at room temperature (25 °C) for fourteen days within the hermetic resealable bags. Sterile water was sprayed onto the specimens to maintain moisture during the incubation period. Fungal colonies and fruiting bodies on the specimens were observed under a Nikon SMZ-1270 stereomicroscope (Nikon Corporation, Tokyo, Japan). Fungal micro-morphological characteristics were observed and photographed using a Nikon ECLIPSE Ni-U compound microscope (Nikon Corporation, Tokyo, Japan), equipped with a Nikon DS-Fi3 camera. Measurements were calculated using PhotoRuler 1.1 (The Genus Inocybe, Hyogo, Japan). Images used for figures were processed with Adobe Photoshop 2021 (Adobe Systems, San Jose, CA, USA). Pure fungal cultures were obtained from single spores following the method described by Chomnunti et al. [23]. Germinated conidia were individually transferred to new potato dextrose agar (PDA) media and cultured at 25 °C in the dark. Potato dextrose agar (PDA) media was prepared as described by Senanayake et al. [24]. The research specimens were deposited in the Fungal Herbarium of Jiangxi Agricultural University (HFJAU), and the pure fungal cultures were deposited in the Jiangxi Agricultural University Culture Collection (JAUCC).

### 2.2. DNA Extraction, PCR Amplification, and Sequencing

Total genomic DNA was extracted from pure fungal cultures using the cetyltrimethylammonium bromide (CTAB) method [25]. DNA amplification was performed by polymerase chain reaction (PCR). The internal transcribed spacers (ITS1-5.8S-ITS2), large subunit (28S, LSU), small subunit (18S, SSU), and RNA polymerase II subunit (*RPB2*) gene regions were selected for analyses using the primer pairs ITS1/ITS4 [26], LR0R/ LR5 [27,28], NS1/NS4 [26], and fRPB2-5F/fRPB2-7cR [29]. The total volume of PCR was 25 μL, containing 9.5 μL of ddH2O, 12.5 μL of 2× Rapid Taq Master Mix (Qingke, Changsha, China), 1 μL of DNA template, and 1 μL of each primer (10 μM). PCR thermal cycles for ITS, LSU, and SSU were performed as the following reaction conditions: initial denaturation at 94 °C for 3 min, followed by 35 cycles of 30 s at 90 °C, 50 s at 55 °C, and 1 min at 72 °C, and a final extension period of 10 min at 72 °C. PCR thermal cycles for *RPB2* were performed as the following reaction conditions: initial denaturation at 95 °C for 5 min, followed by 35 cycles of 1 min at 95 °C, 2 min at 52 °C, and 90 s at 72 °C, and a final extension period of 10 min at 72 °C. The purification and sequencing of PCR products were carried out by Tsingke Biotechnology Co., Ltd. (Changsha, China).

### 2.3. Phylogenetic Analyses

The obtained sequences were initially analyzed with related taxa determined by BLASTn search in the GenBank nucleotide database (https://www.ncbi.nlm.nih.gov/, accessed 8 September 2024). Additionally, some other sequences were obtained from relevant publications [12,20,30,31,32]. Single-gene sequences were aligned using the online version of MAFFT 7 [33]. The alignments were checked and edited using BioEdit 7.2.6 [34]. Multi-gene sequences were concatenated using PhyloSuite 1.2.2 [35]. Sequences derived from this study were deposited in GenBank (Table 1 and Table 2).

Maximum likelihood (ML) and Bayesian inference (BI) were used to assess phylogenetic relationships. Maximum likelihood (ML) analysis was performed with RAxML 8.2.10 [36] using a GTR-GAMMA substitution model with rapid bootstrap analysis followed by 1000 bootstrap iterations to estimate ML bootstrap (BS) values. Bayesian inference (BI) analysis was conducted with MrBayes 3.2.6 on PhyloSuite 1.2.2 under partitioned models [35,37]. The best-fit substitution models of DNA evolution for the combined dataset were inferred according to Akaike information criterion (AIC) implemented in ModelFinder on PhyloSuite 1.2.2 [35,38]. In the first analysis of Pleurotheciaceae, the best-fit model was GTR+I+G for ITS, LSU and *RPB2*, and HKY+F+G4 for SSU. In the second analysis, the best-fit model was SYM+I+G4 for ITS, GTR+F+G4 for LSU, and GTR+F+I+G4 for *RPB2*. The datasets were run for 2 million generations, with four simultaneous Markov chains and trees saved every 1000th generation. The first 25% of saved trees were discarded as burn-in. Phylogenetic trees were visualized with FigTree 1.4.4 [39] and edited with Adobe Illustrator 2021 (Adobe Systems, San Jose, CA, USA).

## 3. Results

### 3.1. Phylogeny

According to sequence alignment analysis, the ITS sequence of *Phaeoisaria filiformis* (JAUCC 7109) shares 99.03% similarity (509/514 bp, with one gap) with *P*. *filiformis* (MFLUCC 18-0214). The ITS sequence of *Pleurothecium lignicola* (JAUCC 7034) shares 90.15% similarity (467/518 bp, with 12 gaps) with *P*. *recurvatum* (CBS 131272). The ITS sequence of *Pseudodactylaria jiangxiensis* (JAUCC 7176) shares 94.30% similarity (430/456 bp, with five gaps) with *Ps*. *brevis* (MFLUCC 16-0032). The ITS sequence of *Ps*. *lignicola* (JAUCC 7032) shares 94.08% similarity (435/468 bp, with three gaps) with *Ps*. *camporesiana* (MFLUCC 16-0032).

The phylogenetic relationships of two Pleurotheciaceae species, *Phaeoisaria filiformis* and *Pleurothecium lignicola*, were assessed in the combined analysis using the ITS, LSU, SSU, and *RPB2* gene regions. The ITS, LSU, SSU, and *RPB2* dataset comprised sequences from seventy-six sequences representing fifteen taxa, including *Dematipyriforma* (four sequences), *Rhexoacrodictys* (three sequences), *Saprodesmium* (one sequence), *Coleodictyospora* (two sequences), *Neomonodictys* (three sequences), *Helicoascotaiwania* (three sequences), *Anapleurothecium* (one sequence), *Phragmocephala* (one sequence), *Melanotrigonum* (one sequence), *Sterigmatobotrys* (three sequences), *Adelosphaeria* (one sequence), *Pleurotheciella* (nine sequences), *Phaeoisaria* (twenty-seven sequences), and *Pleurothecium* (fifteen sequences). *Conioscypha minutispora* CBS 137253 and *Conioscypha tenebrosa* GZCC 19-0217 were used as outgroup. The aligned sequence matrix for the combined analysis consists of ITS (735 characters), LSU (910 characters), SSU (1679 characters), and *RPB2* (926 characters), with a total of 4250 characters including gaps. The phylogenetic trees obtained from RAxML and Bayesian inference analyses were essentially similar in topology. The best-scoring RAxML tree is shown in Figure 1. Phylogenetic analyses indicate that the new *Phaeoisaria filiformis strains* (JAUCC 7109 and JAUCC 7110) cluster with ex-type strain of *P*. *filiformis* (MFLUCC 18-0214) in a highly supported monophyletic clade (99% ML BS/1.0 PP). Moreover, *P*. *synnematicus* (NFCCI 4479) is sister to the *P*. *filiformis* clade with moderate ML bootstrap support values (77% ML BS) and Bayesian posterior probabilities (0.96 PP). All species of *Phaeoisari* form a strongly supported monophyletic clade (100% ML BS, 1.0 PP) in the phylogenetic tree. The strains of *Pleurothecium lignicola* form a distinct lineage sister to a clade containing *Pl*. *recurvatum*, *Pl*. *semifecundum* and *Pl*. *aseptatum*, *Pl*. *floriforme* and *Pl*. *pulneyense* (97% ML BS, 1.0 PP). *Pl*. *aquisubtropicum* forms a separate clade, with weak support within Pleurotheciaceae.

The phylogenetic relationships of two Pseudodactylariaceae species, *Pseudodactylaria jiangxiensis* and *Ps. lignicola*, were assessed in the combined analysis using ITS, LSU, and *RPB2* gene regions of fifty-three sequences in Pseudodactylariaceae (sixteen sequences) and related families Chaetomiaceae (four sequences), Lasiophaeriaceae (two sequences), Sordariaceae (three sequences), Chaetosphaeriaceae (four sequences), Helminthosphaeriaceae (three sequences), Boliniaceae (three sequences), Coniochaetaceae (three sequences), Cordanaceae (three sequences), Cephalothecaceae (four sequences), Meliolaceae (five sequences), and Phyllachoraceae (two sequences). *Arthrinium arundinis* CBS 133509 was used as outgroup. The combined dataset of ITS (790 characters), LSU (958 characters), and *RPB2* (973 characters) comprised 2721 characters with gaps. The RAxML and Bayesian inference analyses gave similar results and agreed with previous studies based on multi-gene analyses [40,41,42]. The best-scoring RAxML tree is shown in Figure 2. In the phylogenetic, three strains of *Pseudodactylaria jiangxiensis* (JAUCC 6196, JAUCC 6198, and JAUCC 7176) cluster together with statistical support of 100% ML BS/1.0 PP and form a sister clade with *Ps*. *brevis* (MFLUCC 16-0032 and MFLUCC 16-0034) with statistical support of 100% ML BS/1.0 PP. The newly obtained strains of *Ps. lignicola* (JAUCC 7032 and JAUCC 7111) clustered together with *Ps*. *camporesiana* and *Ps*. *fusiformis*, which are phylogenetically distinct.

### 3.2. Taxonomy

#### 3.2.1. *Phaeoisaria filiformis* D.F. Bao, Z.L. Luo, K.D. Hyde & H.Y. Su, Fungal Divers 99: 114 (2019) (Figure 3)

Fungal Names number: FN 555671.

*Saprobic* on submerged decaying wood in freshwater habitat. Asexual morph: *Colonies* on the decaying wood superficial, effuse, visible as solitary, hairy, dark brown to black, covered with white conidial masses at upper part. *Mycelium* immersed, composed of brown hyphae. *Synnemata* 282.5–678 × 8–15 μm ($\bar{x}$ = 447 × 12 μm, n = 10), erect, straight or slightly flexuous, smooth, dark brown to black, composed of compact and parallel conidiophores, with flared conidiogenous cells in the above half. *Conidiophores* 1–2.5 μm wide, macronematous, synematous, straight or slightly bent, septate, smooth, branched or unbranched, brown to dark drown, paler towards the apex. *Conidiogenous cells* 6–21.5 × 1.5–2.5 μm ($\bar{x}$ = 11.5 × 2 μm, n = 10), polyblastic, discrete, terminal and intercalary, sympodial, cylindrical or tapering towards the tip, hyaline to pale brown, each with one to several cylindrical denticles. *Conidia* 6–9 × 2.5–3.5 μm ($\bar{x}$ = 7.5 × 3 μm, n = 30), acropleurogenous, obovoid to ellipsoidal, base tapered, apex rounded, straight or sinuous, smooth, hyaline, aseptate, guttulate. Sexual morph: see Luo et al. [5].

Culture characteristics: Conidium germinated on PDA within 24 h from single-spore isolation and germ tubes produced from both ends. Colonies on PDA attaining 30 mm in 60 days at 25 °C in dark, circular, flat, rough, margin crenated, with slightly raised, gray mycelium at the middle region, dark brown at the inner ring, paler towards the margin; in reverse, black, brown at the edge.

Material examined: China, Jiangxi Province, Ganzhou City, Yudu Country, Da Shu Bei Village, 25°99′76 N, 115°39′63 E, on submerged decaying wood in a freshwater stream, 12 June 2024, W.M. He, L. Huang, yd-1-1 (HFJAU10500), living culture JAUCC 7109; China, Jiangxi Province, Ganzhou City, Yudu Country, Da Shu Bei, 25°99′85 N, 115°39′83 E, on submerged decaying wood in a freshwater stream, 13 June 2024, W.M. He, L. Huang, yd-2-8 (HFJAU10501), living culture JAUCC 7110.

*Notes:* Phylogenetic analysis shows that our new collections (JAUCC 7109 and JAUCC 7110) clade with the ex-type strain of *Phaeoisaria filiformis* (MFLUCC 18-0214) with a high statistical support (99% ML BS/1.0 PP) (Figure. 1). Comparison of the ITS, LSU, and SSU gene regions reveals 99.03% (509/514 bp, with one gap), 99% (790/798 bp, with three gaps), and 99.79% (963/965 bp, with one gap) sequence similarity between our new collection (JAUCC 7109) and *P*. *filiformis* (MFLUCC 18-0214), respectively. Based on the molecular evidence, we consider they are the same species. *P*. *filiformis* was introduced as a sexual morph on submerged decaying wood from freshwater in Thailand [5]. In this study, we identify our new collection as asexual morph of *P*. *filiformis* phylogenetically and report a new record of this species in China.

In the phylogenetic tree, *P*. *filiformis* (MFLUCC 18-0214, JAUCC 7109, JAUCC 7110) clades to *P*. synnematicus (NFCCI 4479) with a moderate statistical support (77% ML BS/0.96 PP) (Figure 1). Morphologically, our new collection is distinguished from *P*. *synnematicus* by shorter synnemata (282.5–678 μm vs. up to 960 μm) [17]. Additionally, conidia of *P*. *synnematicus* are dimorphic, clavate to ellipsoidal, cylindrical to falcate, base narrowly truncate, tip obtuse, aseptate to 1-septate, while *P*. *filiformis* conidia are obovoid to ellipsoidal, base tapered, apex rounded, straight or sinuous [17].
Figure 3*Phaeoisaria filiformis* (HFJAU10500). (**A**) Conidiomata synnemata on wood. (**B**,**C**) Conidiophores. (**D**–**G**) Conidiogenous cells and conidia. (**H**–**L**) Conidia. (**M**) Germinated conidium. (**N**,**O**) Colony on PAD from above and below. Scale bars: (**A**) 200 μm; (**B**,**C**) 20 μm; (**D**,**E**,**H**,**M**) 10 μm; (**F**,**G**,**I**–**L**) 4 μm.
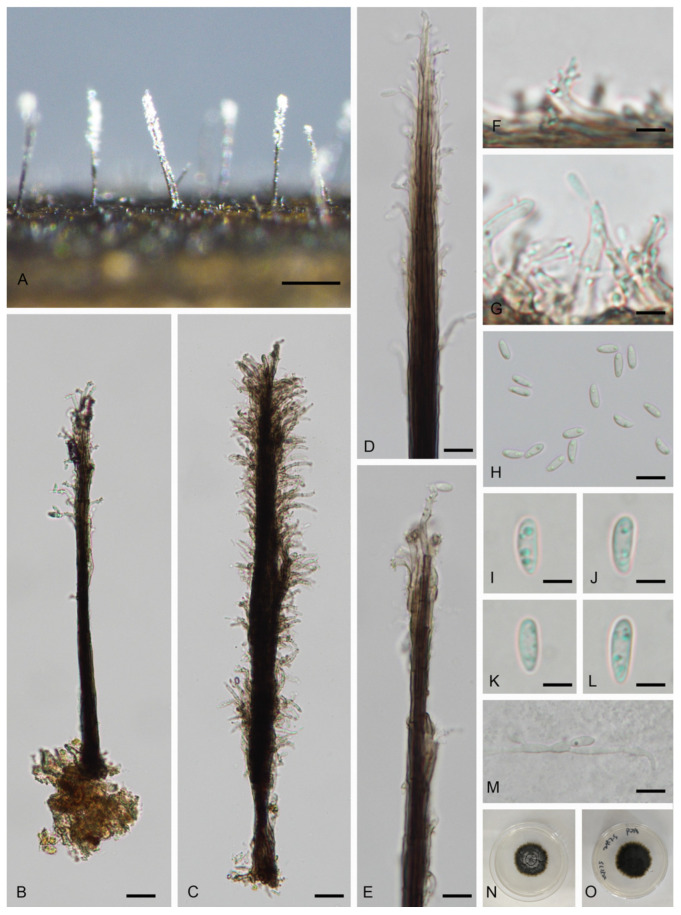



#### 3.2.2. *Pleurothecium lignicola* C.Y. Xu, H.Y. Song & D.M. Hu, sp. nov. (Figure 4)

Fungal Names number: FN 572120.

Etymology: *lignicola* (Latin), referring to growing on wood.

*Saprobic* on submerged decaying wood in freshwater habitat. Asexual morph: *Colonies* on the decaying wood superficial, effuse, soliatary, black, with white conidial masses aggregated at apex. *Mycelium* partly immersed, partly superficial, consist of branched, septate, smooth-walled, brown hyphae. *Conidiophores* 154–280 × 4.5–5.5 μm ($\bar{x}$ = 217.5 × 5 μm, n = 10), macronematous, mononematous, erect, straight or slightly sinuous, mostly unbranched, rarely branched, septate, smooth, cylindrical, dark brown, becoming pale brown to subhyaline towards the apex. *Conidiogenous cells* 13.5–34.5 × 3.5–5 μm ($\bar{x}$ = 24 × 4 μm, n = 10), holoblastic, polyblastic, terminal, integrated, smooth-walled, cylindrical to tapered, pale brown to subhyaline, sympodial, with narrow, cylindrical denticles, denticles 2–3.5 × 1–1.5 μm ($\bar{x}$ = 2.5 × 1.5 μm, n = 15). *Conidia* 14–21.5 × 4.5–6.5 μm ($\bar{x}$ = 17.5 × 6 μm, n = 20), acrogenous, solitary, straight or slightly sinuous, ellipsoidal to clavate, rounded at apex, obtuse to tapered towards base, 3-septate, hyaline, guttulate. Sexual morph: undetermined.

Culture characteristics: Conidium germinated on PDA within 24 h from single-spore isolation. Colonies on PDA reaching 27 mm diameter in 60 days at 25 °C in dark, irregular, raised, edge undulate, dry, with dense, dark brown mycelium on the surface; in reverse, black, becoming dark brown towards the edge.

Materials examined: China, Jiangxi Province, Nanchang City, Xinjian District, Taiping Town, Shenlong Pond Summerhouse, 28°78′23′′56 N, 115°70′69′′61 E, on submerged decaying wood in a freshwater stream, 19 March 2023, C.Y. Xu, L.Y. Liao, W.L. Xia, slt-1-8 (HFJAU10437, holotype), ex-type living culture JAUCC 7034; China, Jiangxi Province, Nanchang City, Xinjian District, Taiping Town, Rhododendron Garden, 28°78′23′′15 N, 115°70′71′′08 E, on submerged decaying wood in a freshwater stream, 10 April 2023, C.Y. Xu, G. Su, Z.H. Jin, djy05 (HFJAU10438), living culture JAUCC 7035.

Notes: In the phylogenetic analysis, *Pleurothecium lignicola* forms a clade with *Pl*. *recurvatum*, *Pl*. *semifecundum* and *Pl*. *aseptatum*, *Pl*. *floriforme* and *Pl*. *pulneyense* with a high bootstrap support (97% ML BS, 1.0 PP) (Figure 1). Morphologically, *Pl*. *lignicola* agrees with the generic concept of *Pleurothecium* in having brown, macronematous conidiophores, polyblastic, sympodially denticulate conidiogenous cells, and hyaline, ellipsoidal or clavate conidia [21,43]. However, *Pl*. *lignicola* differs from *Pl*. *aseptatum* in having larger (154–280 × 4.5–5.5 μm) brown conidiophores and larger (14–21.5 × 4.5–6.5 μm) 3-septate conidia, whereas *Pl*. *aseptatum* has smaller (46–59 × 1.5–2.5 μm) hyaline conidiophores and smaller (8.5–10 × 2–3 μm) aseptate conidia [32]. *Pl*. *recurvatum* differs from *Pl*. *lignicola* in having larger (299–371 μm long, 6–10 μm wide) conidiophores and larger conidia (25–31 μm long, 7–9 μm wide) [5]. Comapared to *Pl*. *floriforme*, *Pl*. *lignicola* has shorter conidiophores (154–280 × 4.5–5.5 μm vs. 190–780 × 3.5–6.5 μm) and smaller conidia (14–21.5 × 4.5–6.5 μm vs. 23–35 × 5.5–9.0 μm) [44]. Thus, our results favor *Pl*. *lignicola* as a new species in the genus.
Figure 4*Pleurothecium lignicola* (HFJAU10437, holotype). (**A**) Colonies on natural substratum. (**B**,**C**) Conidiophores and conidia. (**D**–**F**) Conidiophores. (**F**,**G**) Conidiogenous cells with conidia. (**G**–**H**) Conidia. (**I**) Germinated conidium. (**J**,**K**) Colony on PAD from above and below. Scale bars: (**A**) 100 μm; (**B**–**I**) 10 μm.
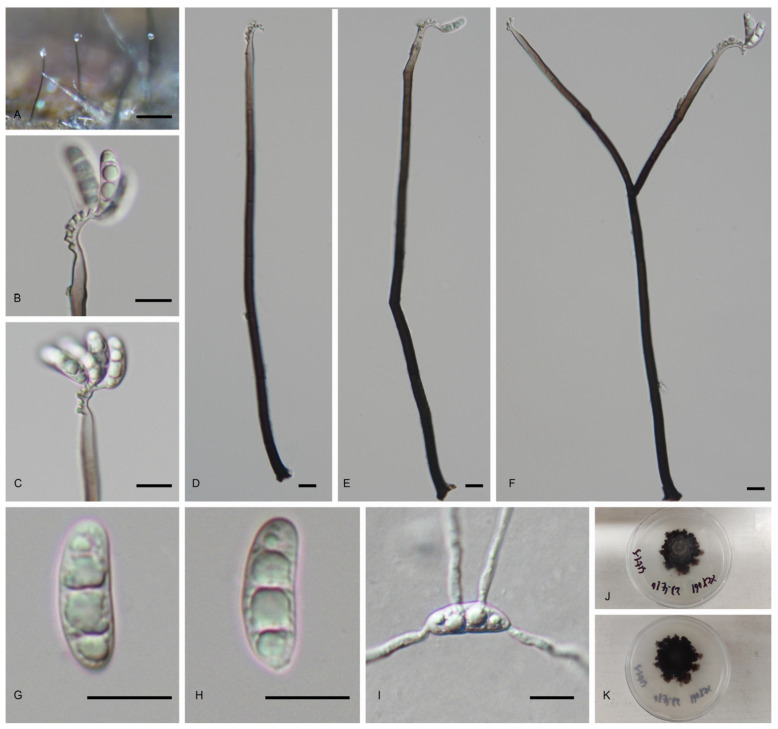



#### 3.2.3. *Pseudodactylaria jiangxiensis* C.Y. Xu, H.Y. Song & D.M. Hu, sp. nov. (Figure 5)

Fungal Names number: FN 572122.

Etymology: Referencing to the collecting location, Jiangxi Province, China.

*Saprobic* on submerged decaying wood in freshwater habitat. Asexual morph: *Colonies* on the substratum superficial, effuse, aggregated, hairy, with white, glistening conidial masses aggregated in heads. *Mycelium* mostly immersed, composed of unbranched, septate, smooth-walled, hyaline hyphae. *Conidiophores* 13–22 × 2.5–3.5 μm ($\bar{x}$ = 16 × 3 μm, n = 10), macronematous, mononematous, erect, straight or slightly curved, unbranched, septate, smooth, thick-walled, cylindrical, sometimes constricted at the septa of the apex, hyaline. *Conidiogenous cells* 4.5–7.5 × 2.5–4 μm ($\bar{x}$ = 5.5 × 3 μm, n = 10), holoblastic, polyblastic, sympodial, terminal, integrated, smooth-walled, cylindrical to tapered, hyaline, denticulate at apical part, denticles cylindrical, flat-topped, 1–2 × 0.5–1 μm ($\bar{x}$ = 1.5 × 0.5 μm, n = 10). *Conidia* 13.5–18 × 2.5–3.5 μm ($\bar{x}$ = 15.5 × 3 μm, n = 20), acropleurogenous, solitary, straight or slightly sinuous, fusiform to clavate, with subobtuse or tapered apex and truncated base, 1-septate, not or slightly constricted at septum, smooth-walled, hyaline, guttulate, lacking a sheath. Sexual morph: undetermined.

Culture characteristics: Conidium germinated on PDA within 24 h from single-spore isolation. Colonies on PDA reaching 40 mm diameter in 60 days at 25 °C in dark, subcircular, margin undulate, uneven, with dense mycelium on the surface, dark brown at the middle region, gray at the inner ring, and brown at the outer ring; in reverse, brown at the middle, paler towards the edge.

Materials examined: China, Jiangxi Province, Ganzhou City, Xingguo County, Yongfeng Town, Xingfu Community, 26°31′97 N, 115°26′61 E, on submerged decaying wood in a freshwater stream, 8 June 2024, W.M. He, L. Huang, xg-1-2 (HFJAU40541, holotype), ex-type living culture JAUCC 7176; China, Jiangxi Province, Nanchang City, Xixia Reservoir, 28°83′20 N, 115°80′83 E, on submerged decaying wood in a freshwater reservoir, 19 May 2023, C.Y. Xu, W.J. Yang, L.Y. Liao, xxsk-1-4 (HFJAU10314), living culture JAUCC 6196; China, Jiangxi Province, Nanchang City, Xixia Reservoir, 28°83′20 N, 115°80′83 E, on submerged decaying wood in a freshwater reservoir, 19 May 2023, C.Y. Xu, W.J. Yang, L.Y. Liao, xxsk-1-7 (HFJAU10316), living culture JAUCC 6198.

Notes: In the phylogenetic tree, *Pseudodactylaria jiangxiensis* clades as a sister taxon to *Ps*. *brevis* with a strong statistical support (100% ML BS/1.0 PP) (Figure 2). The ITS and LSU sequences of *Ps*. *jiangxiensis* (JAUCC 7176) differ from *Ps*. *brevis* (MFLUCC 16-0032) by 26 bp (430/456 bp, with five gaps) and 2 bp (820/822 bp), respectively. Morphologically, *Ps*. *jiangxiensis* resembles *Ps*. *brevis* in having macronematous, mononematous, hyaline conidiophores, polyblastic, integrated, sympodial conidiogenous cells with denticles, and fusiform to clavate, hyaline conidia with one median septum [41]. However, *Ps*. *jiangxiensis* differs from *Ps*. *brevis* in having unbranched conidiophores, smaller conidiogenous cells (4.5–7.5 × 2.5–4 μm), and larger conidia (13.5–18 × 2.5–3.5 μm), whereas *Ps*. *brevis* have branched conidiophores, larger conidiogenous cells (11–25 × 2–5 μm), and smaller conidia (11.5–17.5 × 2.5–4 μm) [41].
Figure 5*Pseudodactylaria jiangxiensis* (HFJAU40541, holotype). (**A**) Colonies on natural substratum. (**B**,**C**) Conidiophores. (**D**) Conidiophore with conidium. (**E**,**F**) Conidiogenous cells. (**G**,**H**) Conidia. (**I**) Germinated conidium. (**J**,**K**) Colony on PAD from above and below. Scale bars: (**A**) 100 μm; (**I**) 10 μm; (**B**–**H**) 2 μm.
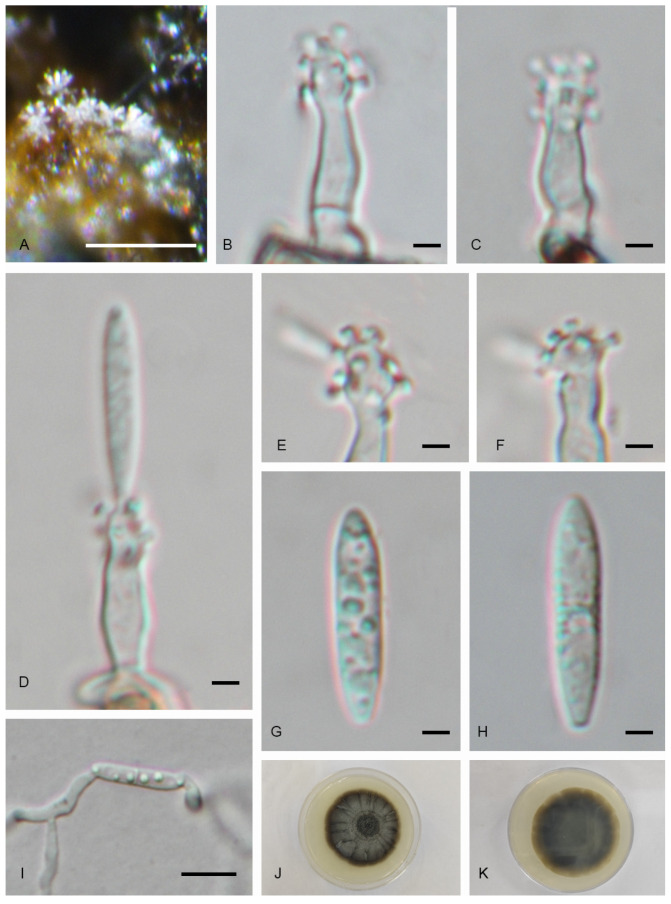



#### 3.2.4. *Pseudodactylaria lignicola* C.Y. Xu, H.Y. Song & D.M. Hu, sp. nov. (Figure 6)

Fungal Names number: FN 572123.

Etymology: *lignicola* (Latin), referring to growing on wood.

*Saprobic* on submerged decaying wood in freshwater habitat. Asexual morph: *Colonies* on the substratum superficial, effuse, gregarious, hairy, with white, glistening conidia aggregated in heads. *Mycelium* partly superficial, partly immersed, consist of septate, smooth-walled, pale brown to hyaline hyphae. *Conidiophores* 27.5–104 μm long, 3.5–7.5 μm wide, macronematous, mononematous, erect, straight or slightly sinuous, unbranched, septate, smooth-walled, cylindrical, sometimes inflated at apex, hyaline. *Conidiogenous cells* 3.5–17 μm long, 3–4.5 μm wide holoblastic, polyblastic, sympodial, terminal, integrated, cylindrical, hyaline, denticulate at the apex, with 2–6 denticles, denticles cylindrical, 1–4 μm long, 1–2 μm wide. *Conidia* 16.5–29.5 × 3.5–5 μm ($\bar{x}$ = 24 × 4.5 μm, n = 30), acrogenous, solitary, straight or slightly sinuous, fusiform, with subobtuse or tapered apex and truncated base, 1–3-septate, not or slightly constricted at the septa smooth-walled, hyaline, guttulate, lacking a sheath. Sexual morph: undetermined.

*Culture characteristics:* Conidium germinated on PDA within 24 h from single-spore isolation and germ tubes produced from both ends. Colonies on PDA reaching 33 mm diameter in 30 days at 25 °C in dark, subcircular, with dense mycelium on the surface, white to pale brown at the middle, brown at the inner ring, and pale brown at the outer ring; in reverse, dark brown at the middle, paler towards entire margin.

Materials examined: China, Jiangxi Province, Ganzhou City, Anyuan County, Sanbai Mountain Cableway, 26°00′55 N, 115°42′33 E, on decaying wood submerged in a freshwater stream, 21 March 2024, W.M. He, L. Huang sbs-1-7 (HFJAU10436, holotype), ex-type living culture JAUCC 7032; China, Jiangxi Province, Ganzhou City, Anyuan County, Sanbai Mountain Dongjiang First Waterfall, 25°00′53 N, 115°27′25 E, on decaying wood submerged in a freshwater stream, 21 March 2024, W.M. He, L. Huang, sbs161 (HFJAU10502), living culture JAUCC 7111.

*Notes:* Phylogenetic analysis shows that *Pseudodactylaria lignicola* forms a strongly supported clade (100% ML BS/1.0 PP) with *Ps*. *camporesiana* and *Ps*. *fusiformis* (Figure 2). Morphologically, *Ps*. *lignicola* has overlapping size of conidiophores (27.5–104 × 3.5–7.5 μm vs. 35–45 × 3.5–5 μm) and conidia (16.5–29.5 × 3.5–5 μm vs. 18–22 × 3.5–4.5 μm) with *Ps*. *camporesiana* [14]. However, *Ps*. *lignicola* can be distinguished from *Ps*. *camporesiana* by having hyaline, conidiophores, and smaller conidiogenous cells (3.5–17 × 3–4.5 μm) with 2–6 denticles and 1–3-septate conidia, whereas *Ps*. *camporesiana* has brown conidiophores with hyaline upper part and larger conidiogenous cells (20–30 × 4–4.6 μm) with a rachis with numerous denticles at apical part and 1-septate conidia [14]. Compared to *Ps*. *fusiformis*, *Ps*. *lignicola* can be distinguished by having conidiogenous cells with 2–6 denticles and 1–3-septate, larger conidia (16.5–29.5 × 3.5–5 μm), whereas *Ps*. *fusiformis* has conidiogenous cells with a rachis with numerous denticles at apical part and 0–1-septate, smaller conidia (11.5–17.5 × 2.5–4.0 μm) [42].
Figure 6*Pseudodactylaria lignicola* (HFJAU10436, holotype). (**A**) Colonies on natural substratum. (**B**–**D**) Conidiophores. (**E**) Conidiogenous cells with attached conidium. (**F**) Conidiogenous cell. (**G**–**I**) Conidia. (**J**) Germinated conidium. (**K**,**L**) Colony on PAD from above and below. Scale bars: (**A**) 100 μm, (**B**–**D**,**G**–**J**) = 5 μm; (**E**,**F**) = 2 μm.
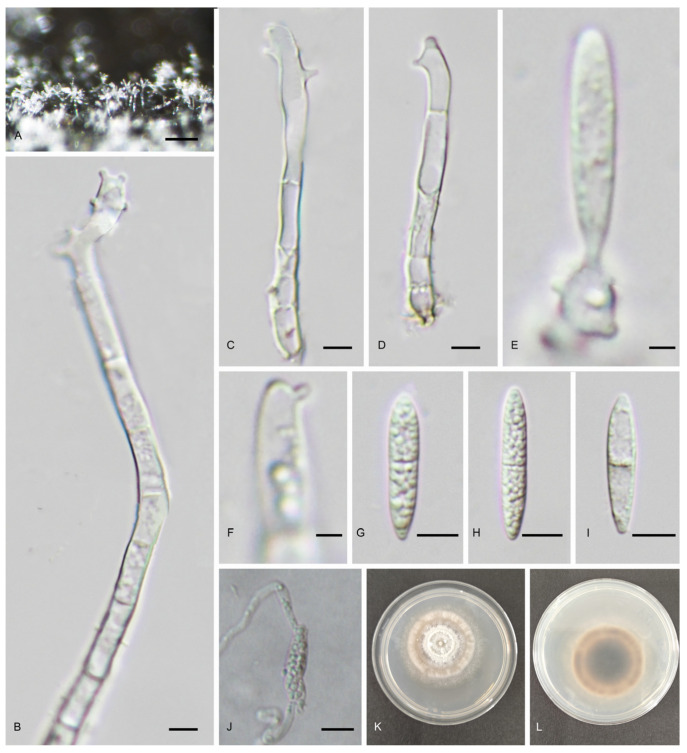



## 4. Discussion

In this study, we introduced three new species, *Pleurothecium lignicola* sp. nov., *Pseudodactylaria jiangxiensis* sp. nov., and *Pseudodactylaria*. *lignicola* sp. nov., and one newly recorded species, *Phaeoisaria filiformis*, from freshwater habitats in China. The discovery of these four new or newly recorded species enriches the freshwater fungal resources in China and further reveals the diversity of fungal morphology. *Phaeoisaria*, *Pleurothecium*, and *Pseudodactylaria* have been reported from both terrestrial and freshwater habitats [5,12,22,31,45,46].

*Phaeoisaria* are relatively common and distributed worldwide, such as in America, Canada, China, Brazil, Germany, Kenya, the Philippines, Thailand, Brunei, Malawi, Malaysia, the Netherlands, South Africa, and India [3,13,17,22,47,48,49]. However, only eight species of *Phaeoisaria* have been recorded in China [30,31,47,50,51]. Furthermore, there are presently only nineteen species of *Phaeoisaria* with molecular data available, whereas some species, such as *P*. *annesophieae*, *P*. *dalbergiae*, and *P*. *fasciculata*, have not been observed to have indeterminate synnemata [8,22,49]. Previously, only one species of *Phaeoisaria* has been identified as having a sexual reproductive pattern, namely, *P*. *filiformis* [5]. In this study, we introduced the asexual morph of *P*. *filiformis* with molecular data provided.

At present, the genus *Pleurothecium* includes one new species introduced in this paper, in addition to eighteen previously known species, eleven of which have been reported in China [3,18,20]. Most species of *Pleurothecium*, including the one new species introduced in this paper, have been collected from aquatic habitats [20]. Furthermore, molecular data are available for only ten species within the genus *Pleurothecium*. *Pl*. *aquisubtropicum* was introduced by Jayawardena et al. [13], which formed a basal branch within *Pleurothecium* with moderate statistical support. Nevertheless, Jayawardena et al. [13] only used molecular data of *Pleurothecium* and *Pleurotheciella* species to conduct phylogenetic analyses. In this study, *Pl*. *aquisubtropicum* does not cluster with other species of the *Pleurothecium*, while forming a separate clade within Pleurotheciaceae with a weak support (Figure 1). Therefore, the taxonomic status of *Pl*. *aquisubtropicum* may require further evidence for confirmation.

Crous et al. [22] introduced the dactylaria-like genus *Pseudodactylaria*, characterized by having erect, hyaline conidiophores, terminal, integrated conidiogenous cells with a denticulate rachis, and hyaline, fusoid-ellipsoid conidia surrounded by a thin mucilaginous sheath. This mucilaginous sheath is absent in *Dactylaria*, representing the distinct difference between the two genera. However, the absence of a mucilaginous sheath in the conidia of some species of *Pseudodactylaria*, including the two species introduced in this study, indicates that the presence or absence of a conidia with mucilaginous sheath is not a diagnostic characteristic to distinguish *Pseudodactylaria* from *Dactylaria* [42]. Apart from *Ps*. *xanthorroeae*, all known species of *Pseudodactylaria* have been collected from aquatic habitats, with seven species found in Thailand, one species from Australia, and only two species reported from China, namely, *Ps*. *hyalotunicata* and *Ps*. *fusiformis* [12,14,17,22,40,41,52]. In this study, we introduced two new species of *Pseudodactylaria* collected from freshwater habitats in China. Until now, there has been no sexual reproductive pattern of *Pseudodactylaria* reported. Therefore, it is likely that species of this genus primarily exist as asexual hyphomycete morphs in natural habitats.

## 5. Conclusions

In this study, we introduced three new species, *Pleurothecium lignicola* sp. nov., *Pseudodactylaria jiangxiensis* sp. nov., and *Pseudodactylaria lignicola* sp. nov., as well as a new record species, *Phaeoisaria filiformis*, from freshwater habitats in Jiangxi Province, China. This achievement was accomplished through rigorous multi-gene phylogenetic analyses and detailed morphological examinations, ensuring the accuracy and robustness of our findings.

The discovery of these new and newly recorded species highlights the rich and underexplored fungal biodiversity present in the freshwater ecosystems of Jiangxi Province. It underscores the importance of conducting comprehensive surveys and studies in such habitats, as they may harbor numerous undescribed species with unique ecological roles and potential applications. Our work contributes to the expansion of the fungal classification system and provides valuable data for future research on the biogeography, ecology, and evolution of freshwater fungi in the region.

## Figures and Tables

**Figure 1 jof-11-00079-f001:**
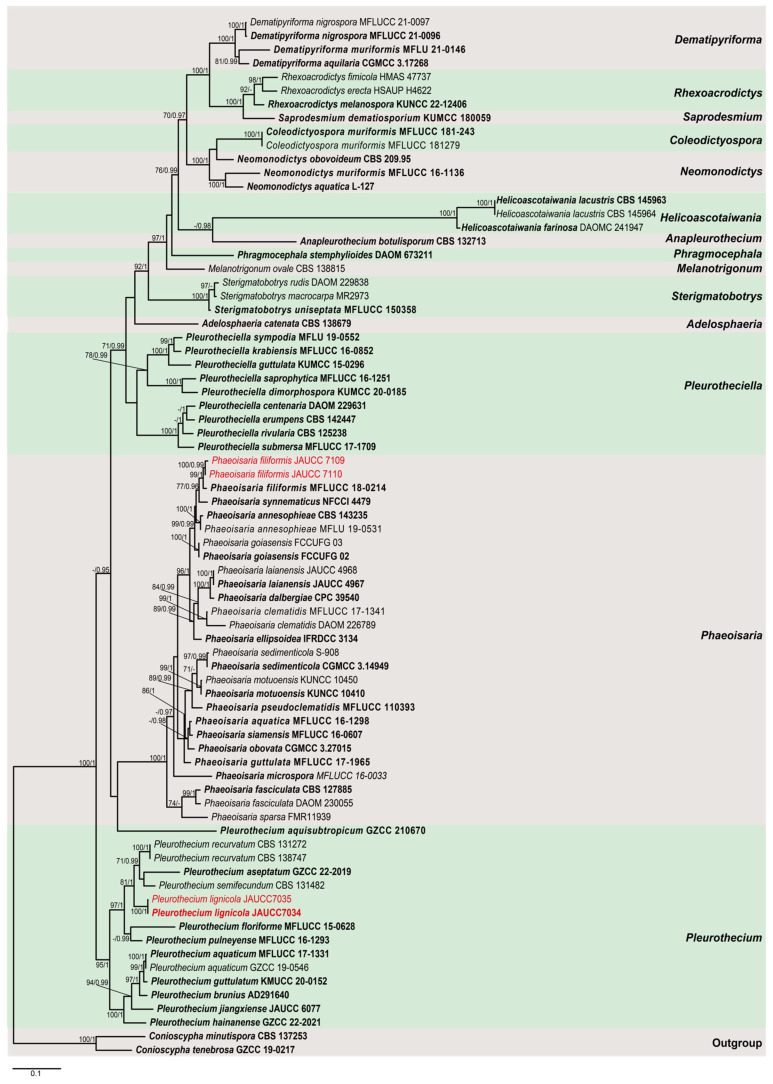
Phylogenetic tree of Pleurotheciaceae generated from maximum likelihood (ML) analysis based on a combined ITS, LSU, SSU, and *RBP2* sequences dataset. Bootstrap support values for maximum likelihood equal to or greater than 70% and Bayesian posterior probabilities equal to or greater than 0.95 are placed near the nodes as ML BS/PP. The tree is rooted with *Conioscypha minutispora* (CBS 137253) and *C*. *tenebrosa* (GZCC 19-0217). Newly generated sequences are indicated in red, and ex-type strains are indicated in bold. Generas are indicated with colored blocks.

**Figure 2 jof-11-00079-f002:**
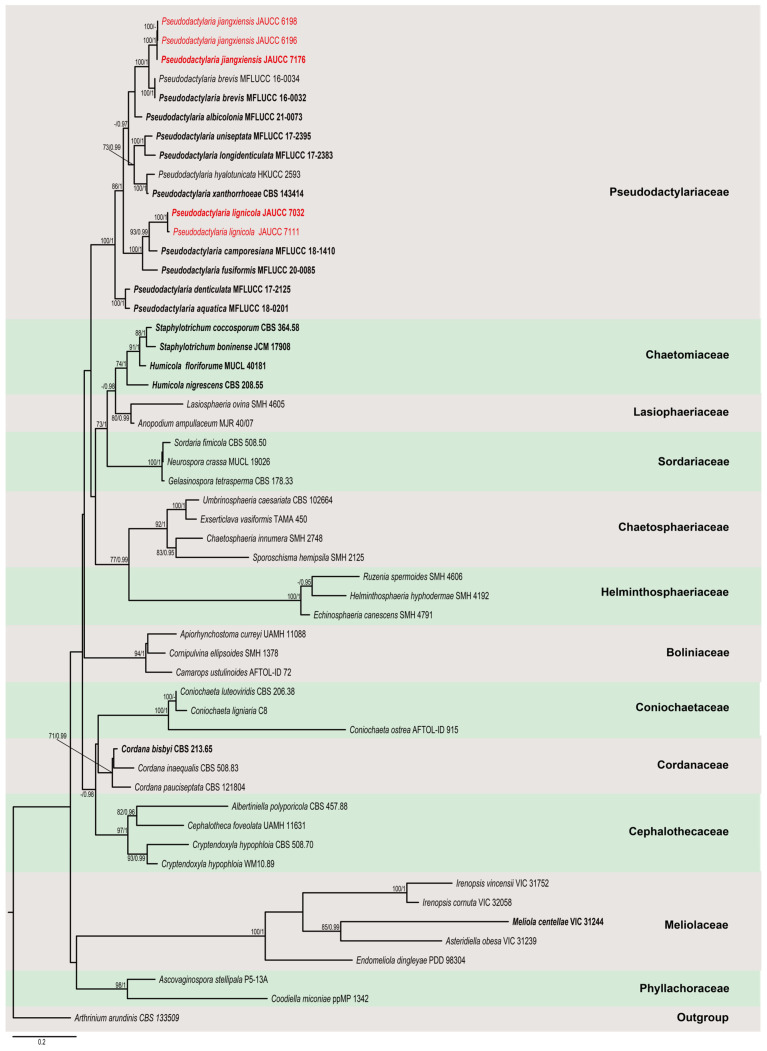
Phylogenetic tree generated from maximum likelihood (ML) analysis based on a combined ITS, LSU, and *RBP2* sequences dataset. Bootstrap support values for maximum likelihood equal to or greater than 70% and Bayesian posterior probabilities equal to or greater than 0.95 are placed near the nodes as ML BS/PP. The tree is rooted with *Arthrinium arundinis* (CBS 133509). Newly generated sequences are indicated in red, and ex-type sequences are indicated in bold. Families are indicated with colored blocks.

**Table 1 jof-11-00079-t001:** List of species, collections, and sequences used in the phylogenetic analyses in Figure 1.

Species	Voucher/Culture	GenBank Accession Number			
		ITS	LSU	SSU	*RPB2*
*Conioscypha minutispora*	CBS 137253 ^T^	KF924559	MH878131	HF937347	N/A
*Conioscypha tenebrosa*	GZCC 19-0217 ^T^	MK804506	MK804508	N/A	MK828514
*Adelosphaeria catenata*	CBS 138679 ^T^	KT278721	KT278707	KT278692	KT278743
*Anapleurothecium botulisporum*	CBS 132713 ^T^	KY853423	KY853483	N/A	N/A
*Coleodictyospora muriformis*	MFLUCC 18-1243 ^T^	MW981642	MW981648	MW981704	N/A
*Coleodictyospora muriformis*	MFLUCC 18-1279	MW981643	MW981649	MW981705	N/A
*Dematipyriforma aquilaria*	CGMCC 3.17268 ^T^	KJ138621	KJ138623	KJ138622	N/A
*Dematipyriforma muriformis*	MFLU 21-0146 ^T^	OM654773	OM654770	N/A	N/A
*Dematipyriforma nigrospora*	MFLUCC 21-0096 ^T^	MZ538524	MZ538558	N/A	N/A
*Dematipyriforma nigrospora*	MFLUCC 21-0097 ^T^	MZ538525	MZ538559	MZ538574	MZ567113
*Helicoascotaiwania farinosa*	DAOMC 241947	JQ429145	JQ429230	N/A	N/A
*Helicoascotaiwania lacustris*	CBS 145963 ^T^	N/A	MN699430	MN699382	MN704304
*Helicoascotaiwania lacustris*	CBS 145964	MN699400	MN699431	MN699383	MN704305
*Melanotrigonum ovale*	CBS 138815	KT278722	KT278711	KT278698	KT278747
*Monotosporella seteosa*	HKUCC 3712	HKU3712	N/A	N/A	N/A
*Neomonodictys aquatica*	L-127 ^T^	MZ686200	OK245417	N/A	N/A
*Neomonodictys muriformis*	MFLUCC 16-1136 ^T^	MN644509	MN644485	N/A	N/A
*Neomonodictys obovoideum*	CBS 209.95 ^T^	EU041784	EU041841	N/A	N/A
*Phaeoisaria aquatica*	MFLUCC 16-1298 ^T^	MF399237	MF399254	N/A	MF401406
*Phaeoisaria annesophieae*	CBS 143235 ^T^	MG022180	MG022159	N/A	N/A
*Phaeoisaria annesophieae*	MFLU 19-0531	MT559109	MT559084	N/A	N/A
*Phaeoisaria clematidis*	DAOM 226789	JQ429155	JQ429231	JQ429243	JQ429262
*Phaeoisaria clematidis*	MFLUCC 17-1341	MF399230	MF399247	MF399216	MF401400
*Phaeoisaria dalbergiae*	CPC 39540 ^T^	OK664703	OK663742	OK663796	OK651159
*Phaeoisaria ellipsoidea*	IFRDCC 3134 ^T^	ON533383	ON533387	N/A	N/A
*Phaeoisaria fasciculata*	CBS 127885 ^T^	KT278719	KT278705	KT278693	KT278741
*Phaeoisaria fasciculata*	DAOM 230055	KT278720	KT278706	KT278694	KT278742
*Phaeoisaria filiformis*	MFLUCC 18-0214 ^T^	MK878381	MK835852	MK834785	N/A
** *Phaeoisaria filiformis* **	**JAUCC 7109**	**PQ443964**	**PQ443976**	**PQ444009**	**N/A**
** *Phaeoisaria filiformis* **	**JAUCC 7110**	**PQ443965**	**PQ443977**	**PQ444010**	**PQ483192**
*Phaeoisaria goiasensis*	FCCUFG 02 ^T^	MT210320	MT375865	N/A	N/A
*Phaeoisaria goiasensis*	FCCUFG 03	MT210321	MT375866	N/A	N/A
*Phaeoisaria guttulata*	MFLUCC 17-1965 ^T^	MG837021	MG837016	MG837026	N/A
*Phaeoisaria laianensis*	JAUCC4967 ^T^	ON937559	ON937557	ON937562	N/A
*Phaeoisaria laianensis*	JAUCC4968	ON937560	ON937561	ON937558	N/A
*Phaeoisaria microspora*	MFLUCC 16-0033 ^T^	MF671987	MF167351	N/A	MF167352
*Phaeoisaria motuoensis*	KUNCC 10410 ^T^	OP626333	OQ947034	OQ947036	N/A
*Phaeoisaria motuoensis*	KUNCC 10450	OQ947032	OQ947035	OQ947037	N/A
*Phaeoisaria obovata*	CGMCC 3.27015 ^T^	PP049488	PP049504	PP049522	PP068788
*Phaeoisaria pseudoclematidis*	MFLUCC 11-0393 ^T^	N/A	KP744501	KP753962	N/A
*Phaeoisaria sedimenticola*	CGMCC 3.14949 ^T^	N/A	JQ031561	N/A	N/A
*Phaeoisaria sedimenticola*	S-908	MK878380	MK835851	N/A	N/A
*Phaeoisaria siamensis*	MFLUCC 16-0607 ^T^	MK607610	MK607613	MK607612	N/A
*Phaeoisaria sparsa*	FMR11939	N/A	HF677185	N/A	N/A
*Phaeoisaria synnematicus*	NFCCI 4479 ^T^	MK391494	MK391492	N/A	N/A
*Pleurothecium aseptatum*	GZCC 22–2019 ^T^	OQ002375	OQ002372	N/A	N/A
*Pleurothecium aquaticum*	MFLUCC 17-1331 ^T^	MF399245	MF399263	N/A	N/A
*Pleurothecium aquaticum*	GZCC 19-0546	MW133897	N/A	MW134679	N/A
*Pleurothecium aquisubtropicum*	GZCC 21-0670 ^T^	OM339436	OM339433	N/A	N/A
*Pleurothecium brunius*	AD291640 ^T^	OQ799373	OQ799347	OQ799346	N/A
*Pleurothecium floriforme*	MFLUCC 15-0628 ^T^	KY697281	KY697277	KY697279	N/A
*Pleurothecium guttulatum*	KMUCC 20-0152 ^T^	MT555415	MT559115	MT559089	N/A
*Pleurothecium hainanense*	GZCC 22-2021 ^T^	OP748934	OP748931	N/A	N/A
*Pleurothecium jiangxiense*	JAUCC 6077	OR853415	OR853420	OR853425	PP078757
** *Pleurothecium lignicola* **	**JAUCC 7034** ** ^T^ **	**PQ443962**	**PQ443974**	**PQ444007**	**PQ569768**
** *Pleurothecium lignicola* **	**JAUCC 7035**	**PQ443963**	**PQ443975**	**PQ444008**	**N/A**
*Pleurothecium pulneyense*	MFLUCC 16-1293 ^T^	N/A	MF399262	MF399228	MF401414
*Pleurothecium recurvatum*	CBS 138747	KT278728	KT278714	KT278703	N/A
*Pleurothecium recurvatum*	CBS 131272	JQ429149	JQ429237	JQ429251	JQ429268
*Pleurothecium semifecundum*	CBS 131482	JQ429158	JQ429239	JQ429253	N/A
*Pleurotheciella centenaria*	DAOM 229631 ^T^	JQ429151	JQ429234	JQ429246	JQ429265
*Pleurotheciella dimorphospora*	KUMCC 20-0185 ^T^	MW981447	MW981445	MW981455	MZ509666
*Pleurotheciella erumpens*	CBS 142447 ^T^	MN699406	MN699435	MN699387	MN704311
*Pleurotheciella guttulata*	KUMCC 15-0296 ^T^	MF399240	MF399257	MF399223	MF401409
*Pleurotheciella krabiensis*	MFLUCC 16-0852 ^T^	MG837018	MG837013	MG837023	N/A
*Pleurotheciella rivularia*	CBS 125238T	JQ429160	JQ429232	JQ429244	JQ429263
*Pleurotheciella saprophytica*	MFLUCC 16-1251 ^T^	MF399241	MF399258	MF399224	MF401410
*Pleurotheciella submersa*	MFLUCC 17-1709 ^T^	MF399243	MF399260	MF399226	MF401412
*Pleurotheciella sympodia*	MFLU 19-0552 ^T^	MT555418	MT559086	MT559094	N/A
*Phragmocephala stemphylioides*	DAOM 673211 ^T^	KT278730	KT278717	N/A	N/A
*Rhexoacrodictys erecta*	HSAUP H4622	KU999964	KX033556	KX033526	N/A
*Rhexoacrodictys fimicola*	HMAS 47737	KU999960	KX033553	KX033522	N/A
*Rhexoacrodictys melanospora*	KUNCC 22-12406 ^T^	OP168085	OP168087	OP168088	OP208807
*Saprodesmium dematiosporium*	KUMCC 18-0059 ^T^	MW981646	MW981647	MW981707	N/A
*Sterigmatobotrys macrocarpa*	MR2973	N/A	GU017317	N/A	N/A
*Sterigmatobotrys rudis*	DAOM 229838	JQ429152	JQ429241	JQ429256	JQ429272
*Sterigmatobotrys uniseptata*	MFLUCC 15-0358 ^T^	MK878379	MK835850	MK834784	N/A

The ex-type cultures are indicated using “T” after strain numbers, and newly generated sequences are indicated in bold. “N/A”stands for no sequence data in GenBank.

**Table 2 jof-11-00079-t002:** List of species, collections, and sequences used in the phylogenetic analyses in Figure 2.

Species	Voucher/Culture	GenBank Accession Number		
		ITS	LSU	*RPB2*
*Albertiniella polyporicola*	CBS 457.88	LT633939	AF096185	LT634061
*Anopodium ampullaceum*	MJR 40/07	N/A	KF557662	N/A
*Apiorhynchostoma curreyi*	UAMH 11088	NR_120207	JX460989	KY931926
*Arthrinium arundinis*	CBS 133509	KF144886	KF144930	N/A
*Ascovaginospora stellipala*	P5-13A	N/A	U85088	N/A
*Asteridiella obesa*	VIC 31239	NR_120256	JX096809	N/A
*Camarops ustulinoides*	AFTOL-ID 72	N/A	DQ470941	DQ470882
*Cephalotheca foveolata*	UAMH 11631	KC408422	KC408398	KC408404
*Chaetosphaeria innumera*	SMH 2748	N/A	AY017375	N/A
*Coccodiella miconiae*	ppMP 1342	MF460365	KX430506	N/A
*Coniochaeta ligniaria*	C8	N/A	AY198388	N/A
*Coniochaeta luteoviridis*	CBS 206.38	NR_154769	FR691987	N/A
*Coniochaeta ostrea*	AFTOL-ID 915	N/A	DQ470959	DQ470909
*Cordana inaequalis*	CBS 508.83	NR_145363	HE672157	N/A
*Cordana pauciseptata*	CBS 121804	HE672149	HE672160	N/A
*Cordana bisbyi*	CBS 213.65 ^T^	N/A	KF746880	N/A
*Cornipulvina ellipsoides*	SMH 1378	N/A	DQ231441	N/A
*Cryptendoxyla hypophloia*	WM10.89	N/A	HQ014708	N/A
*Cryptendoxyla hypophloia*	CBS 508.70	MH859822	NG_058720	N/A
*Echinosphaeria canescens*	SMH 4791	N/A	AY436403	N/A
*Endomeliola dingleyae*	PDD 98304	GU138865	GU138866	N/A
*Exserticlava vasiformis*	TAMA 450	N/A	AB753846	N/A
*Gelasinospora tetrasperma*	CBS 178.33	NR_077163	DQ470980	DQ470932
*Helminthosphaeria hyphodermae*	SMH 4192	N/A	AY346284	N/A
*Humicola floriforume*	MUCL 40181 ^T^	N/A	AF286402	N/A
*Humicola nigrescens*	CBS 208.55 ^T^	AB625592	AB625579	N/A
*Irenopsis cornuta*	VIC 32058	N/A	KC618642	N/A
*Lasiosphaeria ovina*	SMH 4605	AY587923	AY436413	AY600284
*Meliola centellae*	VIC 31244 ^T^	NR_137799	JQ734545	N/A
*Neurospora crassa*	MUCL 19026	N/A	AF286411	N/A
*Pseudodactylaria albicolonia*	MFLUCC 21-0073 ^T^	MW751848	MZ493341	N/A
*Pseudodactylaria aquatica*	MFLUCC 18-0201 ^T^	MZ412510	MZ412522	N/A
*Pseudodactylaria brevis*	MFLUCC 16-0032 ^T^	MH262308	MH262310	N/A
*Pseudodactylaria brevis*	MFLUCC 16-0034	MH262309	MH262311	N/A
*Pseudodactylaria camporesiana*	MFLUCC 18-1410 ^T^	MN796325	MN796326	N/A
*Pseudodactylaria denticulata*	MFLUCC 17-2125 ^T^	OP377887	OP377973	N/A
*Pseudodactylaria fusiformis*	MFLUCC 20-0085 ^T^	MT184905	MT184906	MT188555
*Pseudodactylaria hyalotunicata*	HKUCC 2593	N/A	EU107298	N/A
** *Pseudodactylaria jiangxiensis* **	**JAUCC 6196**	**PQ443959**	**PQ443971**	**PQ483192**
** *Pseudodactylaria jiangxiensis* **	**JAUCC 6198**	**PQ443960**	**PQ443972**	**N/A**
** *Pseudodactylaria jiangxiensis* **	**JAUCC 7176 ^T^**	**PQ555508**	**PQ555509**	**N/A**
** *Pseudodactylaria lignicola* **	**JAUCC 7032 ^T^**	**PQ443961**	**PQ443973**	**PQ483193**
** *Pseudodactylaria lignicola* **	**JAUCC 7111**	**PQ443966**	**PQ443978**	**N/A**
*Pseudodactylaria longidenticulata*	MFLUCC 17-2383 ^T^	OP377857	OP377942	OP473102
*Pseudodactylaria uniseptata*	MFLUCC 17-2395 ^T^	OP377892	OP377978	OP473122
*Pseudodactylaria xanthorrhoeae*	CBS 143414 ^T^	MG386064	MG386117	N/A
*Ruzenia spermoides*	SMH 4606	N/A	AY436422	N/A
*Sordaria fimicola*	CBS 508.50	AY681188	AY681160	N/A
*Sporoschisma hemipsila*	SMH 2125	N/A	AY346292	N/A
*Staphylotrichum boninense*	JCM 17908 ^T^	NR_137527	AB625568	N/A
*Staphylotrichum coccosporum*	CBS 364.58 ^T^	MH857813	MH869345	N/A
*Umbrinosphaeria caesariata*	CBS 102664	N/A	AF261069	N/A

The ex-type cultures are indicated using “T” after strain numbers, and newly generated sequences are indicated in bold. “N/A” stands for no sequence data in GenBank.

## Data Availability

The data generated from this study can be found in the TreeBASE (http://purl.org/phylo/treebase/phylows/study/TB2:S31896?x-access-code=d84d1cedbf40fad1e8d3d50c4a8c979f&format=html, accessed on 15 December 2024).

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
