# Peer review of "Four New or Newly Recorded Species from Freshwater Habitats in Jiangxi Province, China"

_jof, 2025, doi:10.3390/jof11010079_

Round 1
Reviewer 1 Report
This study analyzes samples collected from freshwater habitats in China as well as the discovery of four newly species by morphology and phylogenetic analysis. I consider that describes a complete experimental study and the text is well prepared.
The manuscript can be accepted for publication, with minor observations.
I think the image quality and format can be significantly improved so that there are not blank spaces between pages 2 and 3.

Author Response
Dear Reviewer,
Thank you very much for your valuable comments and suggestions on our manuscript. We have carefully considered each point you raised and have made the following revisions to improve the quality of our paper. Here are the chanages I made in response to your insightful comments. Thank you again for your valuable feedback.
Comments 1: At line 2, The title proposed by the authors is “Four new or newly recorded species from freshwater habitats, China” and is acceptable, however, I think the wording could be improved, in order to make it clear that they were collected in some region of China.
Response 1: Thank you for your comment regarding the title. I agree that specifying the region in China where the species were collected would improve clarity. Based on your suggestion, I have revised the title to "Four new or newly recorded species from freshwater habitats in Jiangxi Province, China." I believe this change better reflects the geographical scope of our study.
Comments 2: At lines 53-57, Regarding figure 1, I think that the quality of the image can be significantly improved, as well as the necessary formatting so that there are no blank spaces between page 2 and 3.
Response 2: Thank you for your feedback on Figure 1. I have made appropriate adjustments to improve the image quality, ensuring it is clearer and more visually appealing. Additionally, I have repositioned Figure 1 to follow the paragraph at line 61, which begins with "Pleurothecium was proposed by von Höhnel [19] and typified by P. recurvatum." This change has effectively minimized the blank space between pages 2 and 3.
Comments 3: At lines 88, 91, In the materials and methods section, specifically in the "collection of specimens", it says that "ziploc bags" are used, which is a trademark. Here either the supplier's information is included or the wording is changed to avoid mentioning trademarks, for example "they were placed in hermetic resealable bags."
Response 3: Thank you for pointing out the use of the trademark "ziploc bags" in the "Collection of Specimens" subsection of the Materials and Methods section. To address this, I have revised the wording at lines 96 and 99, replacing "ziplock bags" with "hermetic resealable bags." This change ensures that the manuscript avoids mentioning trademarks while accurately describing the materials used.
Comments 4: At line 105, Please move the subtitle to page 5.
Response 4: Thank you for your suggestion regarding the placement of the subtitle on line 105. I have made the appropriate adjustments to ensure that the subtitle and its corresponding content are now together on page 4. The revised subtitle is now on line 114.
Comments 5: At lines 202-206, Regarding Figure 2, I believe that the quality of the image can be significantly improved, as well as the necessary format can be provided so that the figure caption is not separated from the image on another page, but rather remains on the same page as the figure.
Response 5: Thank you for your feedback on Figure 2. I have made adjustments to improve the image quality, ensuring it is clearer and more visually appealing. Additionally, I have modified the formatting of Figure 2 so that the figure caption remains on the same page as the image, as per your suggestion. This change enhances the readability and coherence of the presentation. The revised Figure 2 now starts on line 210.
Comments 6: At line 396, Please spell the word “discussion” correctly.
Response 5: Thank you for pointing out the spelling error at line 396. I apologize for my carelessness. I have corrected the word "discussion," and it is now spelled correctly at line 408. Thank you for your attention to detail.
Comments 7: At line 438, The “Conclusion” section is missing, please write it after the “Discussion” section.
Response 6: Thank you for your suggestion regarding the missing "Conclusion" section. Following your advice, I have added the "Conclusion" section after the "Discussion" section, starting at line 451. I truly appreciate your guidance in improving the manuscript. If you have any other comments or suggestions, please feel free to share them. I am eager to continue refining the paper based on your valuable feedback.
The added “Conclusion” section reads as follows:
In this study, we introduced three new species, Pleurothecium lignicola sp. nov., Pseudodactylaria jiangxiensis sp. nov., and Pseudodactylaria lignicola sp. nov., as well as a new record species, Phaeoisaria filiformis from freshwater freshwater habitats in Jiangxi Province, China. This achievement was accomplished through rigorous multi-gene phylogenetic analyses and detailed morphological examinations, ensuring the accuracy and robustness of our findings.
The discovery of these new and newly recorded species highlights the rich and underexplored fungal biodiversity present in the freshwater ecosystems of Jiangxi Province. It underscores the importance of conducting comprehensive surveys and studies in such habitats, as they may harbor numerous undescribed species with unique ecological roles and potential applications. Our work contributes to the expan-sion of the fungal classification system and provides valuable data for future research on the biogeography, ecology, and evolution of freshwater fungi in the region.
I appreciate your constructive comments and suggestions throughout the review process. They have been invaluable in improving the quality of our manuscript. If you have any further comments or recommendations, please feel free to share. Thank you again for your time and expertise.
Thank you and best regards.
Yours sincerely,
Chenyu Xu
Corresponding author:
Name: Dian-Ming Hu
E-mail: Dianminghu1@163.com
Reviewer 2 Report
The article is dedicated to the study of freshwater fungi. The authors isolated, identified and characterised four strains of freshwater fungi. These strains belonged to the genera Phaeoisaria, Pleurothecium and Pseudodactylaria. Among them, new species were identified.
1) At the end of the introduction, I recommend writing the purpose of the study and briefly outlining the approaches and methods used in the work.
2) The authors have described the 'Materials and methods' and 'Results' sections in detail. There are no comments.
3) In the 'Discussion' section, add information about which bioactive compounds are produced by the strains of the identified genera. This is because you stated in the introduction that 'In addition, freshwater fungi have important application potential in the production of a variety of bioactive substances [8-10]'.
4) The list of references is not organised according to the rules of the journal. It should be corrected.
Overall conclusion: I recommend to accept the article after minor revisions.
The article is dedicated to the study of freshwater fungi. The authors isolated, identified and characterised four strains of freshwater fungi. These strains belonged to the genera Phaeoisaria, Pleurothecium and Pseudodactylaria. Among them, new species were identified.
1) At the end of the introduction, I recommend writing the purpose of the study and briefly outlining the approaches and methods used in the work.
2) The authors have described the 'Materials and methods' and 'Results' sections in detail. There are no comments.
3) In the 'Discussion' section, add information about which bioactive compounds are produced by the strains of the identified genera. This is because you stated in the introduction that 'In addition, freshwater fungi have important application potential in the production of a variety of bioactive substances [8-10]'.
4) The list of references is not organised according to the rules of the journal. It should be corrected.
Overall conclusion: I recommend to accept the article after minor revisions.
Author Response
Dear Reviewer,
Thank you very much for your valuable comments and suggestions on our manuscript. We have carefully considered each point you raised and have made the following revisions to improve the quality of our paper. Here are the changes I made in response to your insightful comments. Thank you again for your valuable feedback.
Comments 1: At the end of the introduction, I recommend writing the purpose of the study and briefly outlining the approaches and methods used in the work.
Response 1: Thank you for your valuable suggestion. Following your advice, I have revised the last paragraph of the introduction to include the purpose of the study and a brief outline of the approaches and methods used. The specific changes are in lines 82 to 92. The revised content is as follows:
In this study, we investigated freshwater fungi from Jiangxi Province, China, with the aim of identifying these fungi and clarifying their systematic placement. The research methodology employed a combination of morphological examination and multi-gene phylogenetic analysis. By meticulously observing the morphological characteristics of the fungal specimens and conducting rigorous phylogenetic analyses based on molecular data, we were able to accurately identify and classify the fungi. Utilizing this approach, we introduced three new species, namely Pleurothecium lignicola sp. nov., Pseudodactylaria jiangxiensis sp. nov., and Pseudodactylaria lignicola sp. nov., as well as a new record species, Phaeoisaria filiformis.
Comments 2: The authors have described the 'Materials and methods' and 'Results' sections in detail. There are no comments.
Response 2: Thank you for your thorough review and positive feedback on the 'Materials and Methods' and 'Results' sections. We appreciate your acknowledgment of the detailed descriptions provided. If you have any further comments or suggestions, please feel free to share.
Comments 3: In the “Discussion” section, add information about which bioactive compounds are produced by the strains of the identified genera. This is because you stated in the introduction that 'In addition, freshwater fungi have important application potential in the production of a variety of bioactive substances [8-10]'.
Response 3: Thank you for your valuable feedback. After careful consideration, I have removed the statement in the introduction that mentioned "freshwater fungi have important application potential in the production of a variety of bioactive substances [8-10]." This decision was made because, currently, there is a lack of research regarding the production of bioactive compounds by the identified genera. As a result, I am unable to add information about which bioactive compounds produced by the strains of the identified genera in the “Discussion” section. The original intent of that statement was to highlight the importance of freshwater fungi. However, based on your insightful suggestion, I agree that it may not be appropriate to include it in the introduction at this stage. I appreciate your guidance and will continue to refine the manuscript accordingly.
Comments 4: The list of references is not organised according to the rules of the journal. It should be corrected.
Response 4: Thank you for pointing out the issue with the reference list. I apologize for not adhering to the journal's formatting rules. I have carefully reviewed the journal's guidelines and corrected the reference list accordingly to ensure it follows the required format.
I appreciate your constructive comments and suggestions throughout the review process. They have been invaluable in improving the quality of our manuscript. If you have any further comments or recommendations, please feel free to share. Thank you again for your time and expertise.
Thank you and best regards.
Yours sincerely,
Chenyu Xu
Corresponding author:
Name: Dian-Ming Hu
E-mail: Dianminghu1@163.com